# Metagenomic Analysis of Duodenal Microbiota Reveals a Potential Biomarker of Dysbiosis in the Course of Obesity and Type 2 Diabetes: A Pilot Study

**DOI:** 10.3390/jcm9020369

**Published:** 2020-01-29

**Authors:** Agnieszka Sroka-Oleksiak, Agata Młodzińska, Małgorzata Bulanda, Dominika Salamon, Piotr Major, Maciej Stanek, Tomasz Gosiewski

**Affiliations:** 1Department of Mycology, Chair of Microbiology, Faculty of Medicine, Jagiellonian University Medical College, 31-121 Krakow, Poland; agnieszka.sroka@uj.edu.pl; 2Department of Molecular Medical Microbiology, Chair of Microbiology, Faculty of Medicine, Jagiellonian University Medical College, 31-121 Krakow, Poland; 3Bioidea Company, 02-796 Warsaw, Poland; 4Department of Infection Epidemiology, Chair of Microbiology, Faculty of Medicine, Jagiellonian University Medical College, 31-121 Krakow, Poland; 52nd Department of General Surgery, Jagiellonian University Medical College, 31-501, Krakow, Poland

**Keywords:** *Bifidobacterium*, duodenal microbiota, next-generation sequencing (NGS), obesity, type 2 diabetes

## Abstract

Numerous scientific studies confirm that, apart from environmental and genetic factors, a significant role is played by gastrointestinal microbiota in the aetiology of type 2 diabetes and obesity. Currently, scientists mainly focus on the distal intestinal microbiota, while the equally important proximal parts of the intestine are overlooked. The aim of the study was a qualitative analysis of the structure of the duodenal mucosa microbiota in groups of patients with obesity and with type 2 diabetes and where obesity qualified for bariatric surgery: sleeve gastrectomy. The microbiological results obtained were compared with some clinical parameters. As a result, it was possible to determine the microbiological core that the treatment and control groups had in common, including phyla: Firmicutes, Proteobacteria, and Actinobacteria. The patients with obesity and with type 2 diabetes and obesity presented a significantly lower number of genus *Bifidobacterium* compared to healthy subjects. Furthermore, the numbers of *Bifidobacterium* were positively correlated with the high density lipoprotein (HDL) concentration in the groups under study. The obtained results indicate that bacteria of the genus *Bifidobacterium* should be considered in the future in the context of a potential biomarker in the progress of type 2 diabetes and obesity.

## 1. Introduction

The number of patients with obesity and type 2 diabetes (T2D) is increasing at an alarming rate in both developed and developing countries. The World Health Organization (WHO) has recognised diabetes and obesity as the first non-infectious epidemic. According to WHO findings from 07.04.2016, over 422 million people in the world have diabetes and every tenth person is obese [1]. It is also worrying that the problem of diabetes and obesity is increasingly affecting children and adolescents. Despite the significant progress in medicine, no effective means of combating these diseases have been found and forecasts for subsequent years predict that by 2040 the number of people with diabetes in the world will exceed 642 million [2].

In the study of relationships between microbiota and obesity, type 2 diabetes, or other metabolic diseases, the main research tool is next-generation sequencing, where the baseline material subjected to the appropriate processing and metagenomic analyses is constituted by bacterial DNA isolates from faecal samples from patients. Therefore, the microbiota of the large intestine is quite well described [3]. Some studies, however, suggest a different approach that assumes that this type of research should focus primarily on the small intestine microbiota [3,4]. In the literature, microbiological analysis of upper parts tract, e.g., duodenum, especially in the progress of obesity or diabetes, is rarely the subject of discussion [3,4].

The main reason for the lack of work on this subject is the difficulty in accessing the proximal sections of the intestine [4] since biopsy collection is associated with an invasive intervention that requires the use of a gastroscope. Other reasons are the acidic pH of the stomach and the presence of digestive enzymes that inhabit the small intestine that significantly reduce the number of microbial cells [5]. Most literature data based on duodenal microbiota studies that can be found mainly concern paediatric patients with celiac disease [6,7,8].

The duodenal section of the gastrointestinal tract is located at the intersection between the stomach, secreting digestive enzymes, and the jejunum and ileum, which absorb nutrients. Due to its strategic position in the gastrointestinal tract, the duodenum has important functions related to the digestive process and the absorption of nutrients [9]. Therefore, the structure of the microbiota within this section is worth investigating.

This study aimed to characterise the structure of the microbiota of the duodenal mucosa in patients with obesity, as well as with type 2 diabetes and obesity, who qualified for bariatric surgery - sleeve gastrectomy.

## 2. Experimental Section

### 2.1. Patients

The study included 66 patients of the 2nd Department of General Surgery, Jagiellonian University Medical College (17 with obesity, 22 with type 2 diabetes and obesity, and 27 healthy patients, with normal body weight (BMI 18.5–24.9 kg/m^2^), who had entered the stomach cancer screening programme. The inclusion criteria for the groups of patients with obesity and with type 2 diabetes (T2D) and obesity were as follows: patients with clinical diagnosis of obesity/diabetes, aged 20 to 70 years, administration of oral drugs for at least 2 years after diagnosis (only for T2D), disease duration of at least 2 years, and BMI >35 kg/m^2^. The exclusion criteria were: age under 20 and over 70 years, antibiotic therapy within 30 days before duodenal biopsy sampling, use of probiotic therapy within the 30 days before duodenal biopsy sampling, confirmed gastrointestinal infections, chronic inflammatory bowel disease, congenital and acquired immune deficiencies, alcohol or drug addiction, conditions requiring psychiatric treatment, pregnant women, patients with LADA (latent autoimmune diabetes of adults) or MODY (maturity onset diabetes of the young), and lack of consent to participate in the study or withdrawal of consent during the study.

#### Ethics Consideration

All subjects gave their informed consent for inclusion before they participated in the study. The study was conducted in accordance with the Declaration of Helsinki and the protocol was approved by the Jagiellonian University Bioethical Committee (No. KBET/47/B/2009).

### 2.2. Samples

The tested materials were biopsies from the descending part of the duodenum, obtained with the use of a gastroscope. The patients underwent routine laboratory testing, including fasting blood glucose, glycosylated haemoglobin (HbA1c), total cholesterol, low density lipoprotein (LDL), high density lipoprotein (HDL), triglycerides, and alanine aminotransferase (ALT). The biological materials obtained, maintained in deep-freeze conditions, together with the test results and patients’ signed consent forms to participate in the study, were delivered to the laboratory of the Department of Molecular Medical Microbiology, Chair of Microbiology, Jagiellonian University Medical College.

Bacterial DNA was isolated from 66 duodenal biopsy samples and 5 sterile water samples (included in the study as a negative control) using Mini Genomic kit (A&A Biotechnology, Gdańsk, Poland) according to the method developed by Gosiewski et al. [10]. The concentration and purity of total DNA isolates were measured with use of NanoDrop (Thermo Scientific).

Due to the physiologically small number of bacteria colonizing the duodenum, it was necessary to use the nested-PCR method, which allowed increasing the sensitivity and specificity of the isolates amplified for sequencing. Primers targeting the 16S rRNA gene V3 and V4 regions, the composition of the reaction mixture, and the amplification programme are presented in Table 1.

The products of amplification were subjected to electrophoresis to verify the size of amplicon. Further stages were prepared according to the protocol for the MiSeq sequencer (Illumina).

The microbiological data obtained at each taxonomic level were correlated with the clinical parameters of patients (total cholesterol, LDL, HDL, TG, ALT, and HbA1c), as well as age and BMI.

### 2.3. Bioinformatics Analysis

The first stage of the bioinformatics analysis included merging paired-end reads and quality analysis. Merged reads with average quality Q <15 (phred +33 quality scores) were removed from further analysis. Next, deduplication and clustering into operational taxonomic units (OTUs) were performed. Sequences were clustered with 97% identity threshold. The created OTUs were then annotated according to a custom 16S sequence database. The custom database (BioMeta16SRef, v1.1.6) contains 19316 16S bacterial and archaeal sequences, received from the Greengenes and NCBI (National Center for Biotechnology Information) database on 10 September 2018. The OTU sequence annotation was performed with the use of the QIIME software (assign_taxonomy.py and summarize_taxa.py scripts) [11]. The annotated reads obtained from the negative controls (water samples, *n* = 5) were removed from diagnostic samples and from further analysis. In the next step, the OTU sequences were aligned to the reference 16S gene sequences (BioMeta16SALN, v1.1.6), containing aligned 16S reference sequences) with the use of the align_seq.py script and the PyNAST algorithm (QIIME).

### 2.4. Statistical Analysis

Patient characteristics: In order to select the appropriate statistical test, the Shapiro–Wilk test was performed to assess the presence or absence of normal distribution. Only the data for the LDL concentration parameter met the test conditions, so an ANOVA test was performed with the identification of differences between the groups by Tukey post-hoc analysis. The non-parametric Kruskal–Wallis test with post-hoc analysis (Nemenyi test) was used to analyse the remaining data.

Alpha biodiversity analysis between OTU frequencies was performed with the use of alpha_rarefaction.py pipeline implemented in the QIIME package. Statistics were calculated with the use of compare_alpha_diversity.py (QIIME) script for each biodiversity index. Statistical analysis was performed with two tests: *t*-parametric and *t*-non-parametric (Monte Carlo).

Beta biodiversity analysis between OTU frequencies was performed with the use of beta_diversity.py pipeline implemented in the QIIME package. The maximum read count for analysis was determined as the ratio of 85% of the number of reads representing the smallest sample (36TK: 8,142; 0.85 × 8142 = 6921). Statistical analysis was based on a two-sided Student’s *t*-test. Nonparametric *p*-values were calculated with 999 Monte Carlo permutation.

Analysis of OTUs, assigned to appropriate taxonomic levels within the two treatment groups and a control group, was performed with the use of summarize_taxa.py program of the QIIME package. Statistical analysis was performed by group_significance.py script (QIIME). Two types of ANOVA and Kruskal–Wallis statistical tests were performed for each taxonomic level. The comparative analysis was carried out in the following subgroups: T2D and control, T2D and obese, as well as obese and control. For this purpose, the non-parametric t-test was used (Monte Carlo, 999 permutations).

Correlation analysis of microbiological and clinical data: Division into individual taxonomic levels was made with the use of summarize_taxa.py script implemented in the QIIME package. Statistical correlation analysis of clinical data was performed with the use of observation_metadata _correlation.py script (QIIME). Two types of correlation were performed for each taxonomic level: Pearson and Spearman. The probability value was estimated based on the Fisher’s Z distribution. Significant statistical results were those whose probability value after FDR (false discovery rate) correction was less than 0.05.

## 3. Results

### 3.1. Characteristics of the Study Population

Clinical data of patients from treatment and control groups are presented in Table 2. There were no statistically significant differences observed among the groups in the case of two parameters: age and concentration of LDL.

In order to facilitate the analysis of the presented results, the following group names were used: T2D—the group of patients with type 2 diabetes and obesity, obese—the group of patients with obesity without diabetes, and control—the group of healthy patients.

### 3.2. Metagenomic Sequencing

After sequencing of 66 diagnostic samples, 9,536,638 reads of the 16S rRNA gene were obtained. The number of paired reads amounted to 8,606,568 (90.25%). The average sizes of the reads were 74,540.2 per sample (the minimal number of reads per sample was 8142, the maximal 206,703, and the median 63,860.5). A phylogenetic summary of the results is presented in Table 3.

The present results were obtained using the Bioidea software called BioMeta16S (https://bioidea.com.pl/en, v1.1.0). The developed by Bioidea Company sequence database, BioMeta16SRef (v1.1.6), was created on the basis of data from many sources—primarily the Greengenes and NCBI databases. Periodic and automatic updates to the BioMeta16SRef database enable continuous enrichment with new microbial sequences. The BioMeta16S package allowed the identification of bacteria with an accuracy of 98.50% to the order level and 95.96% to the genus level, as shown in Table 3.

At the species level, this value was much lower, i.e., 39.61%, due to many sequences that showed a high (at 97%) but the same percentage of similarity. This problem is inevitable when short fragments are sequenced. Here, usually the first result is automatically selected, which does not always reflect the correct procedure. For this reason, only OTUs that were explicitly assigned to a particular species were included in the L7 level analysis.

Alpha diversity, expressed in the Chao1 index in the control and T2D groups, was similar and slightly higher in the group of obese patients, as seen in Figure 1a. However, these differences were not statistically significant (*p* > 0.05) as other parameters, such as Shannon index, phylogenetic diversity (PD) whole tree, and observed OTUs, as shown in Figure 1b–d.

Beta diversity (weighted and unweighted UniFrac) was also similar between the treatment and control groups, Figure 2a, b. In each of the analysed groups, both parametric and nonparametric test values were similar and not statistically significant (*p* > 0.05).

Due to a large number of taxa from level L3 to level L7, the description contained in the text refers only to a few selected statistically significant observations. Those taxa whose percentage in all 3 examined groups was lower than 1% were assigned to the category other.

All sequences generated from the isolates of duodenal biopsy specimens were assigned to 7 phylogenetic phyla, as seen in Figure 3. In both the treatment group and control group, bacteria belonging to two phyla—Firmicutes and Proteobacteria—were predominant. After those, a significant relative percentage belonged to the phylum Actinobacteria. Other types were found in much smaller or trace numbers.

After data normalization, the above bacterial phyla were compared in the analysed study groups, as shown in Figure 1. Statistically significant differences between the obese and control groups were observed in relation to the phyla Actinobacteria (*p*_FDR_ = 0.019) and Bacteroidetes (p_FDR_ = 0.007). In the T2D group, the percentage of bacteria belonging to the phylum Bacteroidetes was statistically significantly higher than in the control group (*p*_FDR_ = 0.015). 

In the T2D group, increased relative abundances of the class Gammaproteobacteria (20.02%) in relation to the obese (6.68%) and control (7.93%) groups were detected. These differences were particularly evident at the order level, where the relative abundance of Enterobacteriales was almost four times higher than in the control group (18.48% vs. 4.86%, *p*_FDR_ = 0.087) and more than six times higher than in the obese group (18.48% vs. 2.83%, *p*_FDR_ = 0.028), as shown in Figure 4.

At the genus level, in the obese group, there was a significantly higher, but statistically insignificant, percentage of the genus *Staphylococcus* in relation to other treatment groups (12.61% vs. 0.87% control and 0.99% T2D with obese). In the T2D group, a similar, but also statistically insignificant, upward trend was observed for bacteria of the genera: *Lactobacillus* (8.62% vs. 0.58% control and 2.61% obese) and *Escherichia* (18.42 % vs. 4.84% control and 2.83% obese). The control group had a distinctive and statistically significant percentage of the genus *Bifidobacterium* (5.47% vs. 0.28% obese and 0.69% T2D), (*p*_FDR_ = 0.009), as seen in Figure 5.

These relationships were also present at higher taxonomic levels (family Bifidobacteriaceae and order Bifidobacteriales: 5.49% vs. 0.29% obesity and 0.69% T2D, *p*_FDR_ = 0.002; class Actinobacteria: 20.31% vs. 11.45% obesity and 14.90% T2D, *p*_FDR_ = 0.057). Further analysis allowed the identification of four species within this type in the control group: *Bifidobacterium animalis* (5.20%), *Bifidobacterium adolescentis* (0.15%), *Bifidobacterium longum* (0.10%), and *Bifidobacterium bifidum* (0.02%).

### 3.3. Correlation Analysis

In the control group, a statistically significant positive correlation between HDL concentration and Lactobacillales (Pearson, r = 0.64, *p*_FDR_ = 0.007) was found and a negative correlation between these bacteria and LDL (Pearson, r = 0.57, *p*_FDR_ = 0.003), Figure 6a,b.

In the group of patients with obesity, the number of *Lactococcus* was positively correlated with HDL concentration (Pearson, r = 0.83, *p*_FDR_ < 0.001), Figure 6c.

In the T2D group, a positive correlation was observed between *Bifidobacterium* bacteria and HDL concentration (Pearson, r = 0.87, *p*_FDR_ < 0.001), Figure 6d.

No statistically significant correlations were observed for the remaining parameters.

## 4. Discussion

In this paper, we focused on selected results from a metagenomic analysis of the duodenal microbiota in diabetes in combination with obesity and obese patients (without diabetes) in relation to healthy people. 

In the analysis of biodiversity (both alpha and beta diversity) in the studied groups, there was a lack of statistically significant differences. These results indicate a substantial similarity of the duodenal microbiota, which may indicate a significant share of common bacteria, forming the core duodenal microbiota.

In the tested material, the potential bacteria, which might be described as the common core of the duodenal microbiota in the healthy and treatment groups, were the phyla: Firmicutes, Proteobacteria and Actinobacteria. In the work by Li et al., definite dominance was observed in relation to the phyla Firmicutes and Proteobacteria. Other phyla were present in small quantities [4], which was consistent with the results presented in this paper. On the other hand, Firmicutes and Actinobacteria were the dominant phyla in the studies by Angelakis et al. in both groups and Proteobacteria constituted a much smaller percentage (9.5% in the control group and 4% in the group of obese patients) [9]. These disproportions may result from small numbers, i.e., 5 people per group. Still, other studies showed that the structure of the microbiota within the same section may differ significantly depending on the type of the collected material. For example, in gastric fluid samples, Firmicutes, Bacteroidetes, and Actinobacteria represented the largest percentage and in the gastric mucosa these were: Proteobacteria and Firmicutes [12]. With regard to the above results, the preponderance of Firmicutes and Proteobacteria in biopsies from the patients’ duodenal mucosa seems to be justified because this proximal part of the small intestine is another segment of the digestive tract, separated from the stomach only by the pylorus. In the remaining parts of the gut, especially in the colon, the abundance of Proteobacteria decreases and is replaced by a significant share of Bacteroidetes [12].

According to the literature data, Proteobacteria are associated with a negative impact on human health, mainly due to the presence of endotoxin in the cell wall, i.e., liposaccharide (LPS) [13]. In our work within this phylum, a high percentage of the class Gammaproteobacteria in the group of T2D patients drew our attention (20.02% vs. 7.93% control and 6.68% obese). Similar results were observed in the paper by Mrozinska et al. [14]. The preponderance of this group may be associated with the intake of metformin, commonly used by patients in the treatment of type 2 diabetes. Several studies have shown that the use of certain drugs, including metformin, is associated with an increase in the number of several types of bacteria belonging to the class Gammaproteobacteria: *Enterobacter, Escherichia, Klebsiella, Citrobacter, Salmonella*, and *Proteus* [15]. These results were consistent with several large observational and intervention studies, as well as with the genus *Escherichia* (18.42% vs. 4.84% control and 2.83% obese) in our work. The results of the above studies enhance the arguments for the theory of bacterial translocation to tissues and, thus, a decrease in their abundance in the large intestine [13]. However, these changes may be reversed as a result of treatment with the probiotic strain of *Bifidobacterium animalis* subsp. *lactis 420* (B420), which in studies on mice, carried out by a team led by Amar, caused a reduction in the adhesion of bacteria from the family Enterobacteriaceae (belonging to the phylum Proteobacteria) to the intestinal mucosa and reduction of translocation. As a result, inflammation of the fat tissue normalized [16]. Equally promising results were obtained in the works by Stenman et al. [17,18]. Application of *Bifidobacterium animalis* subsp. *lactis* led to weight loss and reduced fat accumulation and also improved glucose tolerance in mice on a fat-rich diet [17,18].

As in the studies mentioned above, limited bacterial adherence to the intestinal mucosa was also confirmed as well as a decrease in LPS [17]. Other studies have observed that the use of metformin, in combination with B420, might bring much better results in the treatment of diabetes, i.e., improved glycaemic control and insulin sensitivity [19].

These results represent a further solid argument in favour of using *Bifidobacterium animalis* for the reduction of metabolic endotoxemia and for the development of a therapeutic strategy based on bacterial translocation control and mucosal dysbiosis in type 2 diabetes, associated with obesity or metabolic syndrome [16]. To date, only one study was been found in the available literature that has conducted human clinical studies on the effects of *Bifidobacterium animalis* ssp. *lactis 420*. After taking B420, an overweight and obese volunteer group exhibited a reduction in body weight and fat compared to the placebo group [18]. On the other hand, in basic research carried out on stool samples from obese, overweight, lean, and anorexic individuals, it was observed that the number of *Bifidobacterium animalis* was significantly higher in the lean group when compared to the obese [20]. Similar results are attributed to the total number of *Bifidobacterium* bacteria in studies in patients with type 2 diabetes [20]. 

The above data are based solely on studies of the colon and its contents. None of the previously published papers highlight these relationships in other sections of the digestive tract. However, these changes can already be observed in the duodenal mucosa. This is indicated by the results obtained in the present work, in which the percentage of *Bifidobacterium* was significantly higher in the control group (5.47%) compared to the percentages in patients with obesity (0.28%) and with type 2 diabetes and obesity (0.69%), as shown in Figure 5. At the species level, this relationship was not statistically significant, which probably indicates that not only *Bifidobacterium animalis*, but also other species within the genus *Bifidobacterium* (e.g., *B. adolescentis, B. longum*, and *B. bifidum*) may jointly exert an effect visible at higher levels. Particularly noteworthy is the demonstration of a positive and strong relationship (r = 0.87, *p* < 0.001), Figure 6d between the genus of *Bifidobacterium* and HDL concentration in the group of patients with type 2 diabetes and obesity. 

In the work by Salamon et al., after sequencing isolates from stool samples, a similar relationship, but one of average severity, was observed in both patients with type 1 (r = 0.46; *p* = 0.03) and type 2 (r = 0, 43; *p* = 0.03) diabetes [21]. These observations confirm the relationship between the genus of *Bifidobacterium* and HDL concentration in animal studies [22,23,24]. The research results collected so far indicate a significant role of the genus *Bifidobacterium* in the functioning of the intestinal microbiota, both in the distal and proximal parts. Moreover, these bacteria can be considered in the context of a potential marker of intestinal dysbiosis. The next step in the future should involve repeating the research with more patients. If the above-described relationships are found again, the range of *Bifidobacterium* optimal for maintaining homeostasis should be determined as a therapeutic goal. 

A similar relationship with respect to HDL, but in patients with obesity was found for unknown species belonging to the genus *Lactococcus* (r = 0.83; *p* < 0.001), Figure 6c. The respective literature data refer primarily to bacteria with probiotic properties, such as *Lactococcus lactis* [25,26,27]. One review cited in many studies involved selected *Lactococcus lactis* strains [27]. For example, based on many years of in vitro observation, two mechanisms of *L. lactis* action have been proposed, consisting of the adhesion and assimilation of cholesterol, which consequently leads to its reduction in the medium. In turn, in in vivo studies conducted in rats, the administration of two strains of *Lactococcus lactis* subsp. *lactis biovar diacetylactis* N7 and *Lactococcus lactis* subsp. *lactis* 527 reduced total cholesterol and serum triglycerides as well as increased the HDL fraction relative to total cholesterol [27]. In addition to the genera *Lactococcus* and *Bifidobacterium* described earlier, *Lactobacillus* bacteria are also classified as probiotics that have a beneficial effect on human health, including maintaining homeostasis in the intestines, inhibiting inflammatory reactions, and translocating bacteria from the intestines. The correlation analysis in the control group of this study showed a positive effect of the order Lactobacillales on HDL concentration and a decrease in these bacteria, along with an increase in LDL concentration, Figure 6a,b. However, in some reports, the results of meta-analyses indicate that bacteria of the genus *Lactobacillus*, in addition to their beneficial effects, also have an impact on weight gain [28], which can be considered in the context of the results obtained in the present work in which an increasing trend of the genus *Lactobacillus* was observed in the group of obese patients (2.61%) and in those with type 2 diabetes and obesity (8.62%) compared to healthy people (0.58%). Equally interesting observations, confirmed so far in an animal model, were observed in relation to the species *Lactobacillus salivarius. L. salivarius* produces bacteriocins, which increase the number of Bacteroidetes and Proteobacteria and reduces the concentration of Actinobacteria [29]. These results may explain the statistically significant increase in Bacteroidetes, the upward trend of Proteobacteria, and, at the same time, a statistically significant decrease in Actinobacteria in the group of patients with obesity as well as with type 2 diabetes and obesity.

The exact mechanism of action of these bacteria in the context of type 2 diabetes and obesity is not fully understood. However, current knowledge and results of our work support the following described path. Undoubtedly, one of the important factors initiating the cascade of changes taking place in the organism is the excessive and long-term use of a diet rich in fats and sugars, which leads to a decrease in the abundance of *Bifidobacterium*, while increasing fasting glucose and total cholesterol [30]. Bacteria from the *Bifidobacterium* genus are involved, among others, in inhibiting the adhesion of pathogens to the intestinal mucosa [31]. In the case of their deficiency, favourable conditions are created for colonization of the intestines by bacteria belonging to the Proteobacteria phylum (represented in this study in the group of patients with T2D by the Gammaproeobacteria class, the Enerobaceriales order, and the genus *Escherichia*) and Bacteroidetes (present in our study in a significantly higher number compared to the control group), which release the aforementioned LPS into the bloodstream. The role of LPS is to induce and intensify the production of pro-inflammatory cytokines, which in turn leads to a weakening of the tightness of the intestinal wall. As a result, bacterial antigens penetrate the bloodstream and organs, initiating subclinical inflammation. As the amount of LPS circulating in the body increases, the concentration of triglycerides and glucose also increases. These effects contribute to the development of obesity and insulin resistance, which later leads to the development of type 2 diabetes [32]. Thus, the decrease in the number of *Bifidobacterium* in the duodenum and the processes described above may be the initiating factor for further pathogenic changes in subsequent parts of the intestine in the progress of these diseases.

A limitation of this study was the small number of patients in the treatment and control groups. Moreover, the study lacked a third treatment group, which would account for patients with type 2 diabetes without obesity. It is difficult to find such patients, especially when it was necessary to meet the inclusion criteria described above.

## 5. Conclusions

This study demonstrated that diabetic and obese patients have common core microbiota in their duodenum at the phylum level, consisting of Firmicutes, Actinobacteria, and Proteobacteria. Within the phylum Actinobacteria, the relative abundance of the genus *Bifidobacterium* was significantly lower in both obese and diabetic groups in relation to the control group. Observing these changes also on higher taxonomic levels may initiate a pathogenic effect that is intensified in later parts of the intestine. Thus, the genus *Bifidobacterium* should be considered in the future in the context of a potential biomarker in the progress of both type 2 diabetes and obesity.

## Figures and Tables

**Figure 1 jcm-09-00369-f001:**
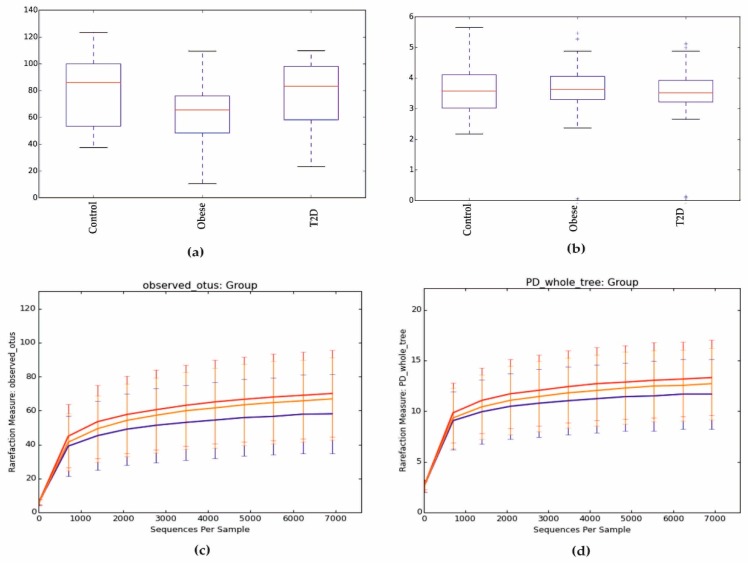
Alpha diversity expressed in: (**a**) Chao1, (**b**) Shannon index, (**c**) observed OUTs, (**d**) PD whole tree. PD: phylogenetic diversity.

**Figure 2 jcm-09-00369-f002:**
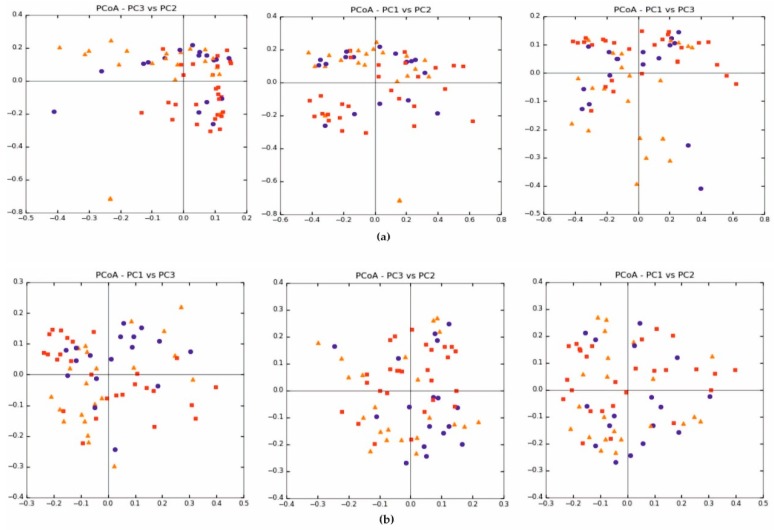
Beta diversity expressed in: (**a**) weighted UniFrac and (**b**) unweighted UniFrac.

**Figure 3 jcm-09-00369-f003:**
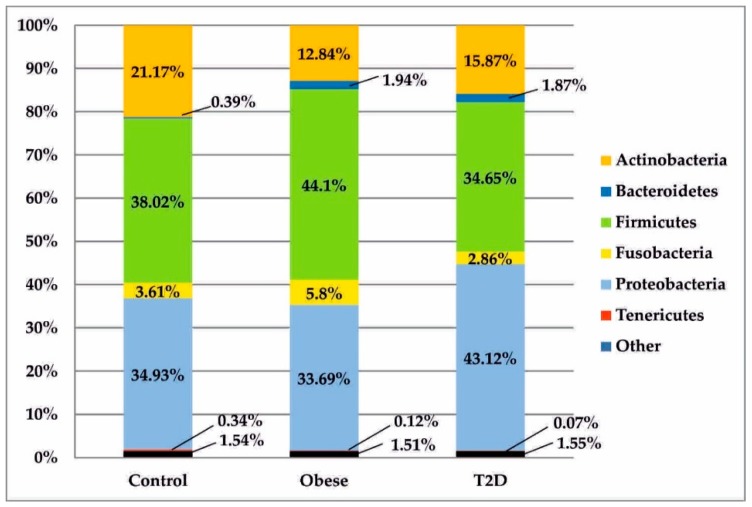
Bacterial profiles in the treatment and control groups at the phylum level. T2D: type 2 diabetes.

**Figure 4 jcm-09-00369-f004:**
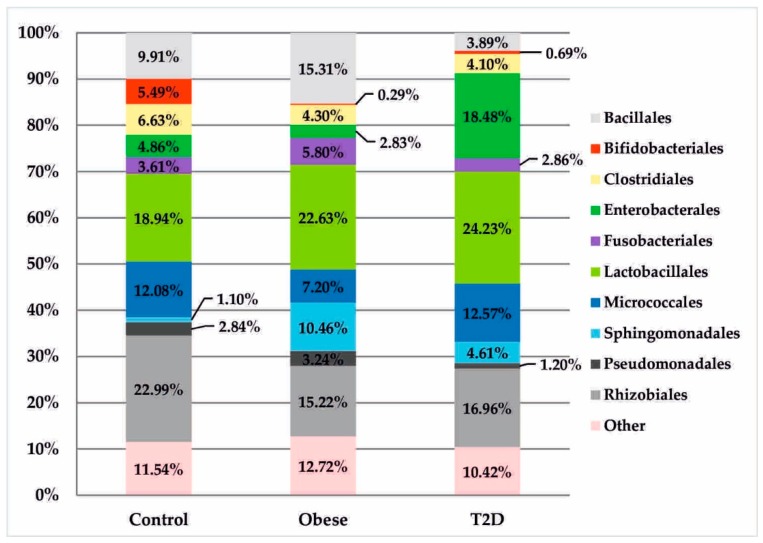
Bacterial profiles in the treatment and control groups at the order level.

**Figure 5 jcm-09-00369-f005:**
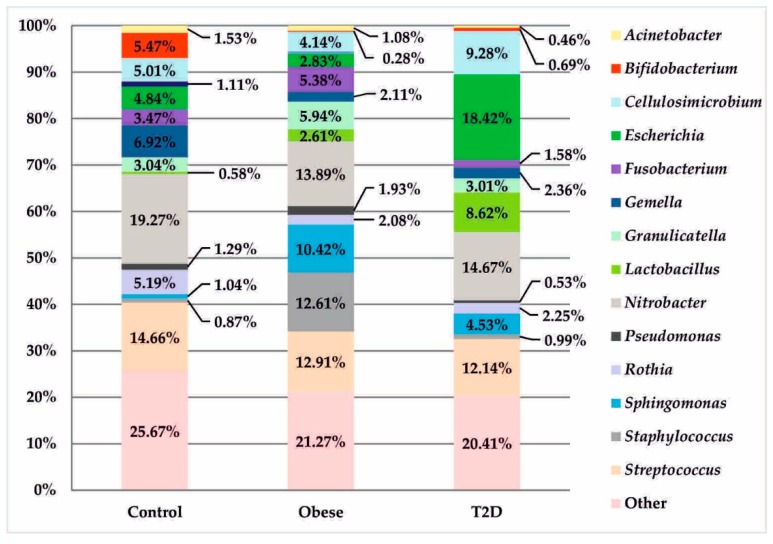
Bacterial profiles in the treatment and control groups at the genus level.

**Figure 6 jcm-09-00369-f006:**
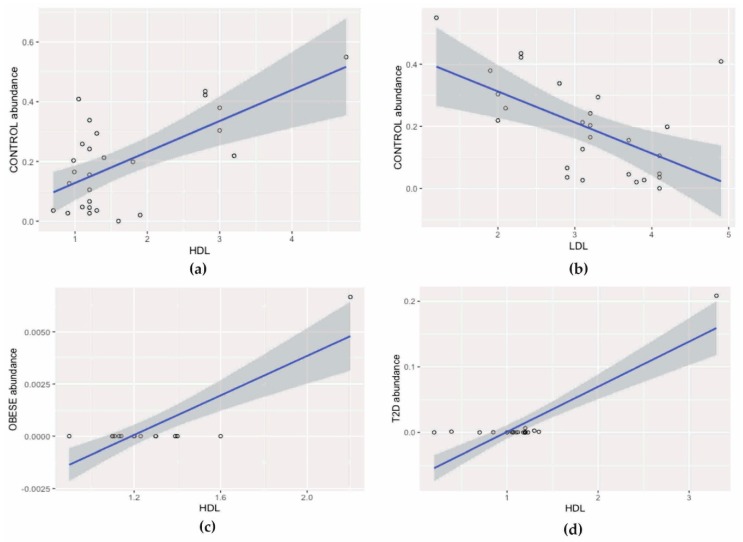
Correlation analysis: (**a**) positive correlation between HDL concentration and Lactobacillales, (**b**) negative correlation between Lactobacillales and LDL, (**c**) positive correlation between HDL concentration and *Lactococcus*, and (**d**) positive correlation between *Bifidobacterium* and HDL concentration. HDL: high density lipoprotein LDL: low density lipoprotein.

**Table 1 jcm-09-00369-t001:** Sequences of primers, reaction mixture, and amplification programme used in the study.

Oligonucleotide Sequence (5′ -> 3′)	Reaction Mixture		Amplification Program
F: ACGGCCNNRACTCCTAC ^1^ R: TTACGGNNTGGACTACHV	Water	2.6 μl	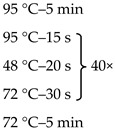
	Kappa	5.0 μl	
	Primer F (10 μM)	0.2 μl	
	Primer R (10 μM)	0.2 μl	
	DNA	2.0 μl	
F: CCTACGGGNGGCWGCAG ^2^ R: GACTACHVGGGTATCTAATCC	Water	10.5 μl	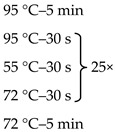
	Kappa	12.5 μl	
	Primer F (10 μM)	0.5 μl	
	Primer F (10 μM)	0.5 μl	
	DNA	1.0 μl	

The overhang adapter sequences (F:TCGTCGGCAGCGTCAGATGTGTATAAGAGACAG and R: GTCTCGTGGGCTCGGAGATGTGTATAAGAGACAG) were added to the internal primer attached to the 5′ end. ^1^ Internal primers. ^2^ External primers.

**Table 2 jcm-09-00369-t002:** Comparative characteristics of clinical data in treatment group and control group.

Parameter	Control (*n* = 27)	Obese (*n* = 17)	T2D (*n* = 22)	*p*-Value
Age [years]	42 (36.0–48.5)	40 (26–42)	45.5 (37.0–55.25)	*p* = 0.179
BMI [kg/m^2^]	23.2 (22.9–23.7)	45 (42.2–5.2)	44 (40.1–47.1)	*p* < 0.001
HbA1c [%]	5.2 (5.1–5.3)	5.3 (5.2–5.5)	6.25 (6.1–6.5)	*p* < 0.001
Total cholesterol [mmol/l]	5.1 (4.9–5.2)	4.5 (3.6–5.0)	3.9 (3.4–5.4)	*p* = 0.003
HDL [mmol/l]	0.98 (0.91–3.0)	1.14 (1.13–1.23)	1 (0.7–1.18)	*p* = 0.040
LDL [mmol/l]	3.16 (0.88)	2.75 (0.65)	2.73 (0.98)	*p* = 0.160
TGs [mmol/l]	0.9 (0.9–1.2)	1.26 (0.9–1.7)	1.6 (1.5–1.9)	*p* = 0.005
ALT [U/l]	20 (18.0–25.6)	44 (28–91)	47 (22.0–178.5)	*p* < 0.001

Data are presented as median (interquartile range) except for concentration of LDL (average). A *p* value of less than 0.05 was considered significant. Abbreviations: T2D (type 2 diabetes), BMI (body mass index), HbA1c (glycated hemoglobin), HDL (high density lipoprotein), LDL (low-density lipoprotein), TGs (triglycerides), ALT (alanine aminotransferase).

**Table 3 jcm-09-00369-t003:** Phylogenetic summary of results obtained.

Taxonomic Level	Abundance ^1^	Number of Reads	Percent of Reads ^2^
kingdom	1	4,844,701	98.50%
phylum	7	4,844,701	98.50%
class	22	4,844,701	98.50%
order	43	4,844,701	98.50%
family	100	4,840,921	98.40%
genus	175	4,720,767	95.96%
species	149	1,948,899	39.61%

The table does not include “other” taxonomies, which indicate an ambiguous result of alignment. ^1^ Number of different taxa assigned to a systematic level. ^2^ Percentage of reads assigned to the appropriate systematic levels.

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
