# Peer review of "Metagenomic Analysis of Duodenal Microbiota Reveals a Potential Biomarker of Dysbiosis in the Course of Obesity and Type 2 Diabetes: A Pilot Study"

_jcm, 2020, doi:10.3390/jcm9020369_

Round 1
Reviewer 1 Report
The study entitled "Metagenomic analysis of duodenal microbiota reveals a potential biomarker of dysbiosis in the course of obesity and type 2 diabetes-a pilot study" aims at qualitatively describing the composition of duodenal microbiota in different groups of patients.
While this kind of studies are not extremely novel for "duodenum focused diseases" such as celiac disease, the authors emphasize the scarcity of information about duodenal microbiota composition in obese and type one diabetes patients.
The manuscript is well written and highlights some interesting differences among the groups of patients, however it could be improved.
1) In the Experimental section the authors should specify which hypervariable regions of the 16s genes they are amplifying.
2) In the Experimental section (lines 76-77) the authors mentioned " before faecal sampling", but no faecal samples have been tested in this study.
3) In the results section authors should show also the figures for alpha and beta diversity although not significant.
Similarly at least genera bar plots or similar graphs should be shown. While the data at phylum level are intriguing, the understanding of which specific bacteria can play a role in the pathogenesis of these metabolic conditions is possible only by analyzing the ranks that are lower than Phylum. The percent of reads at genus level was 95.6% so the data are worth to be shown to back up what is stated in the results section.
4)The discussion gives a nice background of past studies. However it would be interesting to have also a point of view of the authors suggesting what these findings mean in the context of obesity and type 2 diabetes. What is the possible influence of these differences? how mechanistically these bacteria (Enriched or decreased can contribute or being associated with the metabolic conditions?)
Reviewer 2 Report
An association of gut dysbiosis with obesity and diabetes has been well established. However, most of the research in this area is mainly focused on microbiome profile of the distal intenstine due to difficulties in accessing the proximal sections. A pilot study conducted by Sroka-Oleksiak et al. group aimed to investigate the microbiological analysis of duodenal mucosa in patients with obesity and T2D with obesity, who were qualified for sleeve gastrectomy. The authors performed metagenomic analysis supplemented with bioinformatic tools to distribute observed microbial populations into different taxonomic groups, which is quite interesting. The study is very well designed, and the results are well discussed. Overall, the data is impressive and worth for publication. However, there are some minor concerns which should be addressed to improve the quality of work.
In result section 3.2, last para: Authors mentioned that T2D group showed increased relative abundance of class Gammaproteobacteria, which is even evident at the order level. Further, down the taxonomic level, different group present differential relative abundance of microbiota. It would be apt to include a bar graph showing these differences. This would make it easier for the readers to understand the complete picture.
Correlation analysis should also be presented with some figure. This would improve the presentation of the results.
